# How Should the Structure of Smart Cities Change to Predict and Overcome a Pandemic?

Jung-Hoon Kim [1] and Joo-Young Kim [2,*]

1   Smart City Research Center, Korea Institute of Civil Engineering and Building Technology, Goyang-si 10223, Korea; kimjunghoon@kict.re.kr
2   Department of Architecture, Sejong University, Seoul 05006, Korea
*   Correspondence: jooyoung@sejong.ac.kr

**Abstract:** A proposed countermeasure to COVID-19 is a robust healthcare system that can respond and identify transmission paths using information technology. This involves the use of smart city services for tracking an infected person. However, during the COVID-19 pandemic, the healthcare system could only provide data on the number of infected people. Additionally, smart city services could respond neither timely nor sequentially. This study proposed a method for timely and sequential responses, through a flexible combination of the healthcare system and smart city services by envisioning a scenario that sequentially grafts the current status of COVID-19 in Korea. The results are the following. First, the COVID-19 outbreak was summarized in the context of the healthcare system and current smart city services. A method by which the latter could respond to the various needs of the former was suggested. Second, recommendations on combining or dismissing certain smart city services, as per the needs of coping with COVID-19, were summarized. Third, smart city services must be utilized only for addressing pandemics, as data from the healthcare system consists of personal information. Therefore, smart city services for responding to COVID-19 must be flexible.

**Keywords:** smart city; smart city structure; COVID-19; healthcare system; smart city flexibility

## 1. Introduction

The outbreak of the COVID-19 pandemic in 2019 [1] changed people's everyday lives throughout the world [2]. One of the major changes was the transformation of urban quarantine systems [3,4]. Due to COVID-19, one can expect future urban management to differ from the existing conventional model. Moreover, it would be nearly impossible to return to the previously "normal" conditions. In other words, one must now embrace the "new normal," which involves routinized quarantine [5].

Cities have always been vulnerable to large-scale outbreaks of disease [6,7]. Over time, urban quarantine systems have been developed using information technology (IT) and scientific countermeasures [8,9]. In the past, quarantine systems operated only in the areas of disease outbreak [10]. However, quarantine systems today have expanded and are inter-connected by IT [11,12]. Additionally, the development of IT has enabled countries to quickly adopt effective countermeasures employed in other countries [13]. However, it is neither cost- nor time-effective to establish a new infrastructure and system for countering a novel disaster, such as a disease outbreak in cities. Hence, it is imperative that the smart city project (a global urban IT project) [14] be further developed into a system that can deal with future large-scale disasters. In other words, it is necessary to redefine the present urban IT service, or the smart city service, as an extensive countermeasure to disasters.

With the use of smart city technology, Korea has established a response system to COVID-19 [15]. This has several implications, which can be summarized into three main points. First, to counter COVID-19, Korea integrated smart city services with medical services for attaining crucial information to prevent the spread of the pandemic [16,17]. Second, the effective smart city services for tracking the spread of COVID-19 include

medical, crime prevention, traffic and environmental services [18,19]. For example, the conventional method of tracking a patient's movement involved various organizations and took several days (first, the Korea Centers for Disease Control and Prevention (KCDCP) questions patients on their paths of movement; second, the KCDCP requests the National Police Agency to confirm the path; third, the National Police Agency requests the mobile phone company to provide data on this path; fourth, this path is analyzed). However, smart city services can sort out this information within 10 min (first, the KCDCP verifies the patient's information; second, the patient's path of movement is confirmed using smart city services) [20,21]. They have been able to transform the government's COVID-19 response system into 28 real-time sub-systems of information [22]. Third, Korea's COVID-19 data (in terms of transmission route and radius) and countermeasures were quickly shared worldwide [23].

However, the smart city system did face certain issues while responding to COVID-19. These can be summarized as follows; first, smart city services did identify the COVID-19 status but could not respond proactively with a time-series analysis for forecasting and preventing the outbreak [24]. Second, while the smart city structure (infrastructure, data and service) was suitable for operating its services, the pre-existing smart city services could not converge to efficiently counter new problems (such as the emergence of the virus). In other words, the smart city structure lacks the flexibility of adding or removing services, as per emergent needs. Thus, smart city services should have a time-series response structure to proactively prevent possible problems and respond on time. Moreover, they should be flexible in adding an essential service or removing an obsolete one, without any difficulty.

## 2. Literature Review

The COVID-19 outbreak revealed certain limitations of the smart city system. To address these, it is important to have an overview of the smart city structure and healthcare system for COVID-19 control.

### 2.1. Smart City Structure

The definition of smart cities varies, depending on the country and implementing body [25,26]. There are various methods of building and administrating a smart city (e.g., engineering, procurement and construction (EPC) and operation and maintenance (O&M)). Nonetheless, the smart city structure has a common definition. Simply put, smart city structures comprise smart city infrastructure, smart city data and smart city services [27,28].

A simplified representation of the smart city structure is presented in Table 1 and visualized in Figure 1.

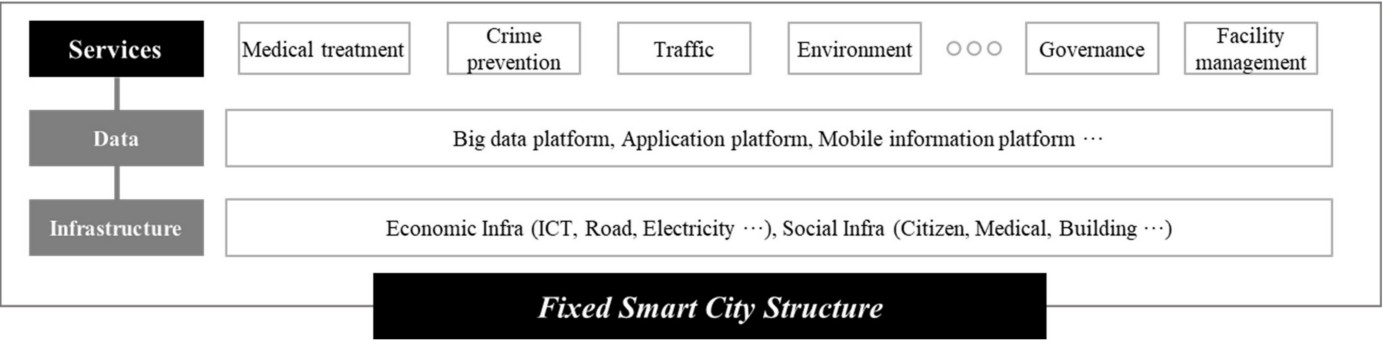

**Figure 1.** Smart city structure.

The stage of designing a smart city structure entails the definition, establishment and management of the processes. Such characteristics impose limitations on the smart structure; for example, they make it difficult to add a new service or remove an obsolete one after a new smart city is built.

**Table 1.** Definition of smart city structure [29].

| Smart City Structure | Definition | Example |
|---|---|---|
| Service | Technology that provides citizens, public institutions and service users with information processed from smart city data | Medical treatment, crime prevention, traffic, environment, governance, facility management, education, culture, logistics, working, living, others |
| Data | Technology of processing the data generated in smart city infrastructure into an optimal form of information for use in a smart city | Big data platform, application platform (e.g., AI, convergence security), mobile information platform |
| Infrastructure | Technology of checking and transmitting urban information essential for smart city services, fundamental facilities necessary for bridging basic communication and electricity | Economic infrastructure (ICT Infra, Urban Infra, etc.), social infrastructure (citizen, medical, building, urban innovation class) |

*2.2. Status of Smart City Services*

Among the components of a smart city structure, smart city services are directly experienced by urban residents or end-users [30–33]. Therefore, they should be specifically reviewed.

2.2.1. Status of Smart Cities around the World

The current conditions of smart city services, in terms of establishment and management, can be outlined as follows. In South Korea, various smart city projects and services have been implemented. In 2018, the smart city service was enacted into a law [34] and defined in 12 smart city service domains [35]. The smart city services in Korea can be outlined as shown in Table 2.

Based on these categories of smart city services, the construction status of each service in Korea [36–38] can be summarized as shown in Table 3. As a representative smart city project in Korea, Pilot city Regulatory Sandbox and Smart City Challenge are in progress. To summarize the smart city services used in this project, medical, crime prevention, transportation and environmental services are the most used.

Table 4 shows the current conditions of smart city service implementation in the USA, Japan, UK, China and Singapore.

The smart city services outlined above were introduced in the respective countries to resolve urban problems. However, they do have some limitations. In response to emergent threats such as COVID-19, these services cannot predict the situation and perform new functions to prevent large-scale transmission within a city. Further, smart city services respond neither timely nor sequentially to disease outbreaks, transmission, prevention and recovery. In other words, it is imperative to investigate smart city services that can counter pandemics.

**Table 2.** The smart city services in Korea.

| Smart City Services | Primary Technology |
|---|---|
| Medical treatment | Healthcare service (comprehensive medical service that combines health service and medical IT), telemedicine |
| Crime prevention | Public safety, monitoring center, CCTV-based tracking, crime prevention response technology |
| Traffic | Cooperative-Intelligent Transport Systems (C-ITS), vehicle-to-everything communication (V2X) |
| Environment | Monitoring environmental pollution, energy consumption, recycling of resources |
| Governance | Participation in decision making, public and social services, interconnecting information and administrative agencies, mobile identification card |
| Facility management | Smart grid, management of declining buildings and infrastructures, remote facility management |
| Education | E-education, offering educational opportunities to low-income families in different regions |
| Culture | Virtual tourism, virtual library, e-shopping, e-commerce, urban cultural assets DB |
| Logistics | Logistics automation, location-based service (LBS) |
| Working | Teleconference, virtual working space |
| Living | Smart building, smart home |
| Others | Digital twin, AI analysis, smart street light, smart pole |

**Table 3.** Status of smart city projects and services.

| Services / Projects | Pilot City | Regulatory Sandbox | Smart City Challenge | Total |
|---|---|---|---|---|
| Medical treatment | 1 | 11 | 1 | 13 |
| Crime prevention | 5 | - | 1 | 6 |
| Traffic | 9 | 2 | 12 | 23 |
| Environment | 7 | 2 | 4 | 13 |
| Governance | 4 | - | 3 | 7 |
| Facility management | 1 | 1 | - | 2 |
| Education | 1 | 1 | - | 2 |
| Culture | | 1 | 3 | 4 |
| Logistics | 1 | - | - | 1 |
| Working | | - | - | |
| Living | 1 | - | | 1 |
| Others | 1 | - | 4 | 5 |

**Table 4.** Progress with smart city services in leading countries.

| Country | Smart City Project | Description |
| --- | --- | --- |
| USA | Smart City Challenge [39] | Establishing a connected transportation network and integrated utilization of shared data, improving public transportation user services, building electric vehicle infrastructure |
| | Federal Smart Cities and Communities Program [40] | Building infrastructure in the city center, revitalization of smart city through sharing of data, new technologies and cases, establishment of smart city evaluation system |
| | IES-City Framework [41] | Establishing a standardized platform for information exchange and cooperation between states |
| | | Nine domains including built environment, water and wastewater, waste, energy, transportation, education, health, socio-economic development, public safety, policing and emergency response system |
| | Smart Grid Program [42] | Promoted by DoE (Department of Energy) and NIST (National Institute of Standards and Technology) |
| | | Develops innovative technology and its methods and guidelines of application |
| | | Applies two-way communication technology for accessing distributed power and energy storage systems of the existing power grid, provides the basic cybersecurity technology |
| Japan | Next-generation energy and social system demonstration projects [43] | Establishment of energy management system (EMS), smart grid, home energy management system (HEMS), building energy management system (BEMS) |
| | Smart Community Demonstration Projects | A project that integrates and manages various energy sources at the regional level for efficient use of energy and creates a comprehensive smart community (e.g., smart transportation system) in line with changes in citizens' lives |
| | Smart city promotion project using data [44] | In 2018, six smart city projects were undertaken (Sapporo City data platform project, Yokohama City scenario project using public and private emergency data, Kakogawa City data-based safe smart town construction project, Takamatsu City data-based smart city construction project, Aizuwakamatsu City citizen-centered smart city project and Saitama City Urawamiso District Data Utilization Project) |

**Table 4.** *Cont.*

| Country | Smart City Project | Description |
|---|---|---|
| | Strategic Innovation Promotion Program [45] | Research in 12 fields including cyberspace-based technology using big data AI, physical-space-based digital data technology, Society 5.0 realization technology using photonic quantum, automatic driving technology |
| UK | Smart London Plan [46] | Efficient convergence of local government, education, healthcare and transportation through digital technology |
| | Smarter London Together [47] | Smart urbanization using urban data and digital technology |
| China | National plan for new Urbanization [48] | Implementation of six policies (distribution of wideband communication networks, informatization of planning and management, smartening of infrastructure, simplification of public service, modernization of industrial development and refinement of local governance) |
| | Pilot projects of IoT and smart city core technology [49] | Research and development in six fields (smart sensing technology and smart terminal, ubiquitous technology and convergence system, urban modeling technology and dynamic sensing system, comprehensive urban decision-making technology and smart service platform, technology and support system for physical convergence of urban information) |
| Singapore | SNPs (Strategic National Projects) [50] | Statewide adoption of digital and smart technologies across Singapore: National Digital Identity, e-payments, Smart Nation Sensor Platform, smart urban mobility, Moments of Life, CODEX (Core Operations, Development Environment, eXchange) |

### 2.2.2. Status of Smart City Service Standardization

Progress in the standardization of smart city services is underway. The process of standardization started at the behest of international organizations specifically for the purpose, such as ITU-T, ISO and IEC. Standardization organizations are further developing and disseminating service standards. Table 5 shows the list of standardization agendas processed by each organization.

**Table 5.** Smart city standardization status.

| Smart City Services | | Description of Standardization Field | Standard |
|---|---|---|---|
| Medical treatment | Smart health | Health information and communications technology (ICT) | ITU-T SG16 [51] |
| | | Health system (health informatics interchange, health-related data and information) | ISO/TC 215 [52] |
| | | Applied healthcare technology and service | TTA TC4 PG419 [53] |
| | | Prevention, management, treatment of disease and tailored health medical service | Digital Health Forum [54] |

The objective behind every organization's standardization is the preemption of smart city services [55]. In particular, private standardization organizations are actively developing standards for elementary technology for smart city infrastructure, data and services [56]. Despite these, standardized smart city services face limitations in adapting to a new environment. This is because smart cities have different service providers, and the data collected by one smart city service need to be integrated with those of other services [57,58]. Complex technical support, such as data form unification and systems connection, is required to provide the necessary emergent services via the existing smart city platform [59,60]. To be specific, the current smart city structure is fixed, and adding a new essential service or removing an outdated one poses challenges [61,62]. In other words, the structure lacks flexibility [63]. Hence, it is crucial to explore a scalable structure of smart city services that can easily add a new essential service and remove an obsolete one.

### 2.3. Status of Pandemic

In order to prevent the spread of COVID-19, countries around the world are implementing various countermeasures, the most common of which involve the healthcare system. The standard process followed by healthcare systems adheres to the phases of the influenza pandemic, as specified by the World Health Organization (WHO) [64], as shown in Figure 2.

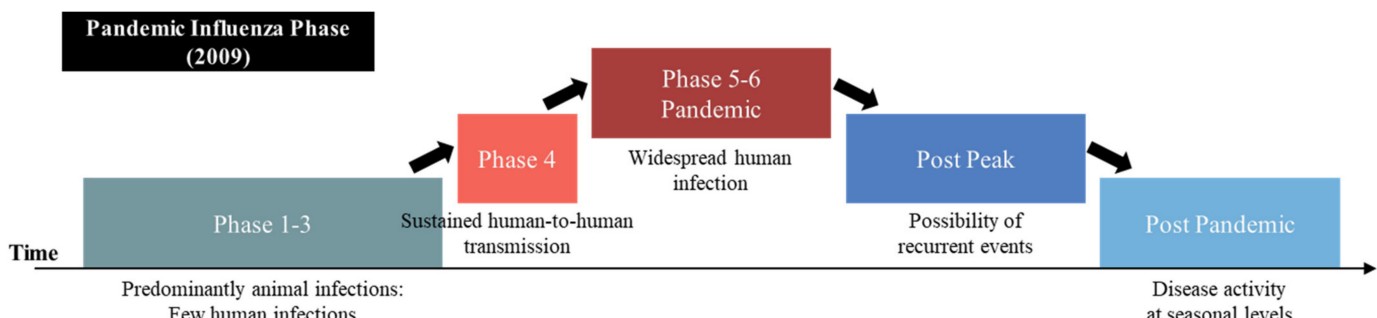

**Figure 2.** WHO phases for the influenza pandemic.

Each phase can be summarized as shown in Table 6.

**Table 6.** Phases for the influenza pandemic.

| Phase | Explanation |
|---|---|
| Phase 1 (Green) | No viruses circulating among animals have been reported to cause infections in humans. |
| Phase 2 (Green) | An animal influenza virus circulating among domesticated or wild animals is known to have caused infection in humans and is, therefore, considered a potential pandemic threat. |
| Phase 3 (Green) | An animal or human–animal influenza re-assorted virus has caused sporadic cases or small clusters of disease in people but has not resulted in human-to-human transmission sufficient to sustain community-level outbreaks. |
| Phase 4 (Orange) | Human-to-human transmission of an animal or human–animal influenza re-assorted virus, able to sustain community-level outbreaks, has been verified. |
| Phase 5 (Red) | This identified virus has caused sustained community-level outbreaks in two or more countries in the same WHO region. |
| Phase 6 (Red) | In addition to the criteria defined in Phase 5, the same virus has caused sustained community-level outbreaks in at least one other country in another WHO region. |
| Post Peak (Dark Blue) | Levels of the influenza pandemic in most countries with adequate surveillance have dropped below peak levels. |
| Post-Pandemic (Blue) | Levels of influenza activity have returned to the levels seen for seasonal influenza, in most countries with adequate surveillance. |

The WHO identifies the phases of pandemics by sharing real-time information on disease outbreaks with major countries such as the United States, EU and Korea, and the completion of a phase, and the move to the next, is determined and implemented on the basis of the increase in each country's affected population.

In the United States, the federal government and state governments respond in different phases [65]. First, the Centers for Disease Control and Prevention (CDC), a federal organization, defined six national pandemic intervals: investigation (green), recognition (yellow), initiation (orange), acceleration (red), deceleration (red) and preparation (green) [66]. Each domain can be divided into the following sub-domains: incident management, surveillance and epidemiology, laboratory, community mitigation, medical care and countermeasures, vaccine, risk communication and state/local coordination. Each interval has a specific, color-coded guideline for easy identification.

Meanwhile, the Korea Centers for Disease Control (KCDC) divided its pandemic response into four phases: concern (blue), notice (yellow), alert (orange) and serious (red) [67] (Table 7).

The phases set by the KCDC escalate or de-escalate on the basis of the number of infections and patients. Hence, it does not require identifying patients' movements or people who had been in close contact with them.

Healthcare systems in all major countries follow time-series analyses. Moreover, the time series is color-coded for easy understanding of non-professionals as well. The time flow of the Korean healthcare system can be visualized as shown in Figure 3

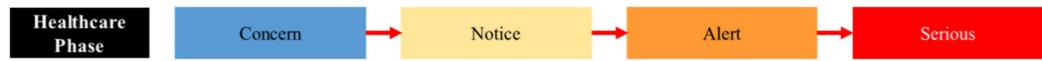

**Figure 3.** Healthcare system phase with time series.

**Table 7.** The KCDC phases according to increase in COVID-19 [67].

| Phase | Type of Crisis | Response |
|---|---|---|
| Concern (Blue) | · Outbreak of a new epidemic abroad<br>· WHO declares Public Health Emergency of International Concern (PHEIC)<br>· Occurrence of diseases with unknown causes, both domestically and abroad | · Monitor the symptoms<br>· Check response plans |
| Notice (Yellow) | · Influx of new infectious diseases from overseas<br>· Limited spread of recurring infectious diseases with unknown cause in Korea | · Operate the cooperation system |
| Alert (Orange) | · Limited domestic spread of new infectious diseases from overseas<br>· Community spread of recurring infectious diseases with unknown cause in Korea | · Operate the response system |
| Serious (Red) | · Community or nationwide spread of new infectious diseases from overseas<br>· Nationwide spread of recurring infectious diseases with unknown cause in Korea | · Mobilize the entire response capacity |

COVID-19 is highly transmissible, especially if one is closer than 2 m (or 6 ft) of a confirmed patient [13]. Therefore, it is crucial to identify the people who were in close contact with confirmed patients for preventing further spread. According to the COVID-19 projections scenario [68], the number of patients will rapidly increase if these identifications are not done on time.

Despite the pressing need for such identifications, the healthcare system only has data on the status of infections and number of confirmed patients. In any case, the healthcare system alone cannot identify the movement paths of confirmed patients and those who were in close contact with them. Thus, it is especially essential that other systems be utilized. Moreover, it is necessary to devise a method that effectively utilizes limited resources, such as disinfection services, along the patient's path of movement.

## 3. Revised Model of Smart City Service Structure

### 3.1. COVID-19 Scenario (Limitations of Current Smart City Structure)

Smart city services use IT to solve urban problems related to the environment, traffic and disasters. However, these services face limitations in responding to large-scale disease outbreaks, such as COVID-19. As mentioned in Section 2.2.2, the primary limitation stems from the lack of flexibility in the smart city structure. To verify this postulate, a COVID-19 scenario, as shown in Figure 4 COVID-19 Scenario: Outbreak of COVID-19 in Korea was assumed, in combination with the healthcare system visualized in Figure 3. Healthcare system phase with time series. The COVID-19 scenario in Figure 2 is based on the window of the COVID-19 outbreak (January 2020–September 2020), number of patients and healthcare phases in Korea [69].

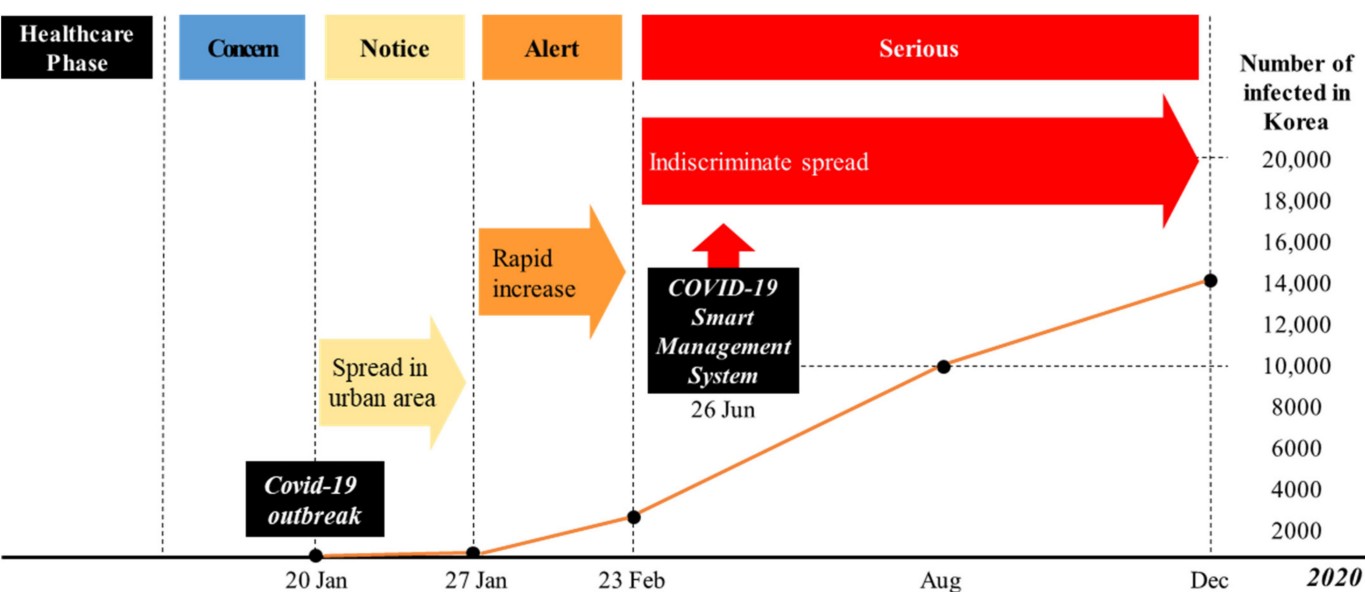

**Figure 4.** COVID-19 Scenario: Outbreak of COVID-19 in Korea.

Figure 4 COVID-19 Scenario: Outbreak of COVID-19 in Korea can be described as Table 8.

**Table 8.** COVID-19 scenario description.

| Phase | Phase Description |
| --- | --- |
| Concern (20 January 2020) | Pre-outbreak of COVID-19; monitoring of suspicious conditions. |
| Notice (20 January~27 January) | First cases of COVID-19 are reported. People in close contact with confirmed patients also get infected, and urban spread begins. |
| Alert (27 January~23 February) | The confirmed patients and their close contacts constitute a rapid increase in the number of infected people, thereby causing an urban emergency. |
| Serious (23 February) | COVID-19 spreads rampantly within the city, and its disease response system reaches its threshold. |

The smart city services summarized in Table 2 are contextualized in the COVID-19 Scenario.

Figure 4 COVID-19 Scenario: Outbreak of COVID-19 in Korea, the former's limitations are revealed, as shown in Figure 5 Outbreak of COVID-19 in Korea and the limitations of smart city services.

The limitations of smart city services can be described as follows (Table 9). During the concern phase, they could not proactively detect the COVID-19 outbreak. During the notice phase, when COVID-19 was spreading, they failed to provide tracking information. During the alert phase, they could not operationalize the quarantine system to arrest the pace of COVID-19 spread in the cities. Finally, during the serious phase, the KCDC introduced the COVID-19 Smart Management System, by aggregating the data on infected persons' paths of movement and operationalizing the quarantine service.

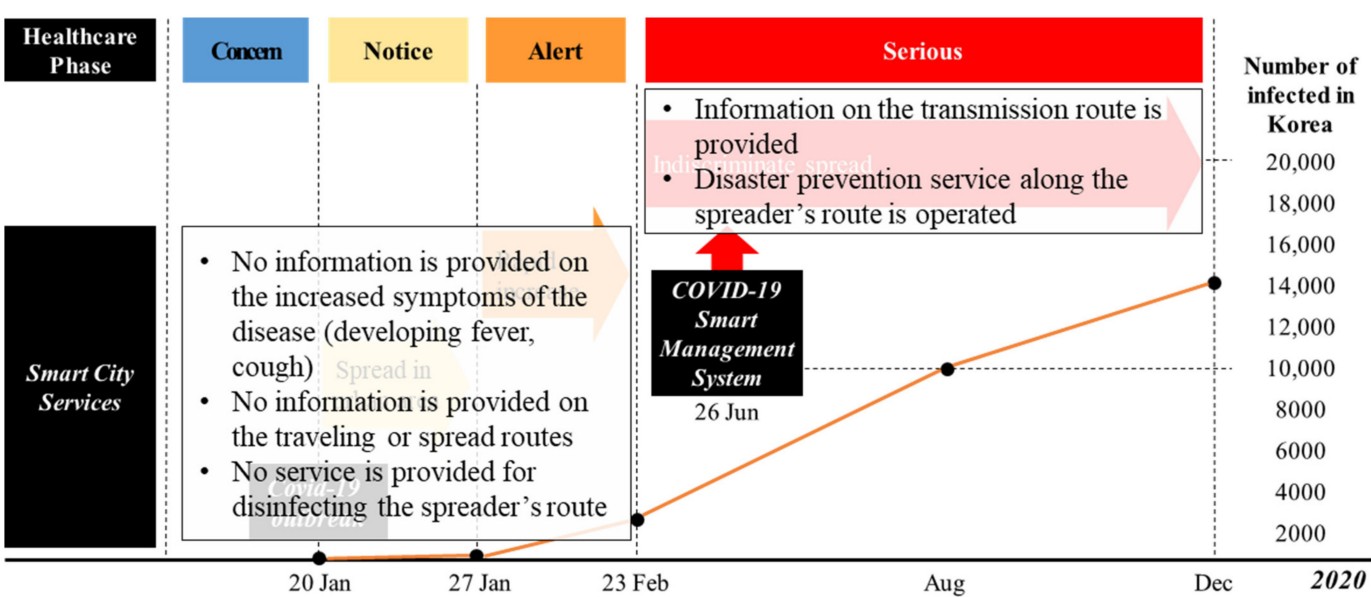

**Figure 5.** Outbreak of COVID-19 in Korea and the limitations of smart city services.

**Table 9.** Limitations of smart city services in each phase.

| Phase | Limitations of Smart City Services |
| --- | --- |
| Concern | No information was provided on detected abnormalities for increased symptoms such as fever and cough. |
| Notice, Alert | No information was provided on the paths of movement of confirmed patients and persons who were in close contact with them.<br>No information was provided on quarantine services along confirmed patients' paths of movement. |
| Serious | Except for the paths of movement of confirmed patients, no other information was provided. |

During the concern, notice and alert phases, smart city services could not operate, as neither the individual smart city services were integrated, nor newly required services established [20,70]. Consequently, time that could have been spent on developing an early response to the COVID-19 outbreak was wasted. It was only during the serious phase (March 2020) that Korea introduced the COVID-19 Smart Management System, which gathered information on confirmed patients' locations, paths of movement and networks for further spread and conducted an epidemiological investigation [71]. This system was integrated with traffic and crime prevention services. It automatically categorized the information in terms of confirmed patients' paths of movement and locations of stay (based on the time spent there) and could thus identify the regional source of infection in areas of a large-scale outbreak (or hotspots).

By launching the COVID-19 Smart Management System, Korea proved that new services could be established by integrating essential functions with existing smart city services [70,72]. However, the system could not serve other functions, except the automatic analysis of confirmed patients' paths of movement [15]. Therefore, the creation of a structure that can respond to infectious diseases even in the earlier phases (concern, notice and alert) remains imperative.

### 3.2. Expansion of Smart City Structure

Smart cities can effectively respond to urban epidemics by controlling the environment and crime prevention services at the smart city center. However, the smart city service structure does not follow a time series, which makes it difficult to combine individual

services for creating a new one. Therefore, smart city services should be converged with healthcare systems that follow a time series. Additionally, by establishing a structure that allows the integration of essential information with smart city services during each phase, a new essential service can be created, such as for countering COVID-19, and discontinued when they become obsolete, as the need may be. As smart services that counteract infectious diseases utilize a large volume of personal information (e.g., information on credit card and public transportation usage), they should be flexibly used only during national crises and discontinued when no longer needed. This can be summarized as in Table 10.

**Table 10.** Smart city services corresponding to each healthcare phase.

| Phase \ Services | Medical Treatment (Infected/Suspected Status Checking) | Crime Prevention (Infected Moving Path Tracking) | Traffic (Infected Moving Path Tracking) | Environment (Contamination Area Disinfection) |
|---|---|---|---|---|
| Concern | Monitor and analyze the number of patients with suspected symptoms | N/A | N/A | N/A |
| Notice, Alert, Serious | · Monitor and analyze the number of confirmed patients <br> · Monitor and analyze suspected symptoms of people who were in close contact with confirmed patients <br> · Provide the location of available hospitals | · Screen feverish people through thermal camera <br> · Track the paths of movement of confirmed patients through CCTV <br> · Monitor and analyze people who had close contact with confirmed patients | · Analyze the confirmed patient's public transportation route <br> · Analyze the public transportation route of people who were in close contact with confirmed patients <br> · Provide information on routes for transporting emergency patients | · Operate the disinfection service along confirmed patients' paths of movement and also along those of people in close contact with confirmed patients |

Figure 6 shows how the expanded structure of the smart city and healthcare system can be applied to the scenario in Figure 4.

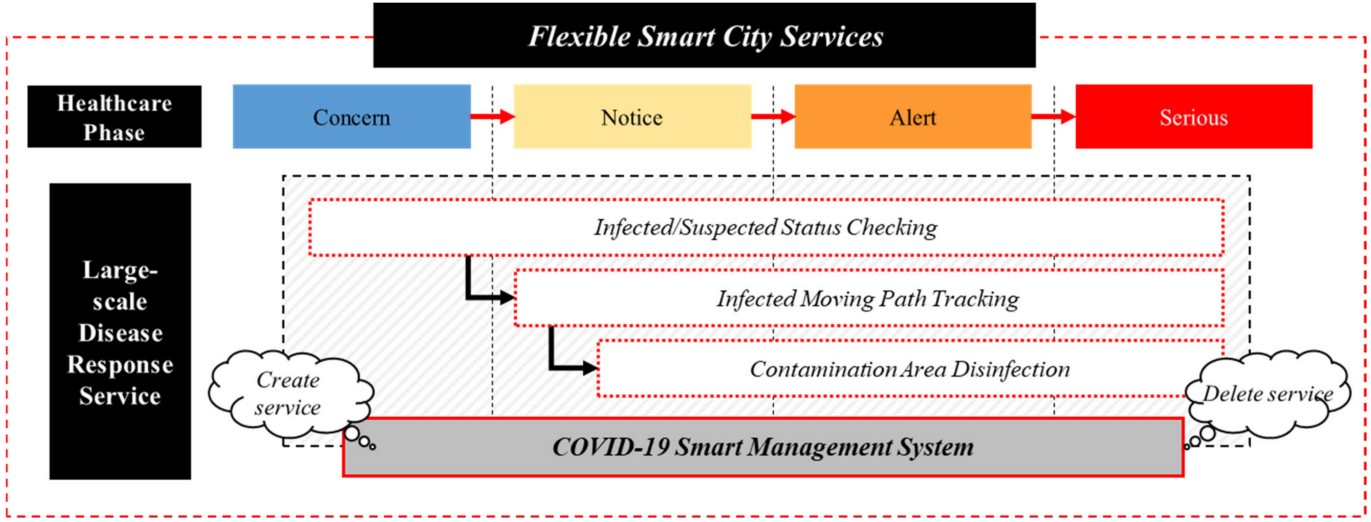

**Figure 6.** COVID-19 scenario with an expanded smart city structure.

Figure 4 COVID-19 Scenario: Outbreak of COVID-19 in Korea.

### 3.3. Smart City Service Structure for Countering Pandemic

By applying the expanded smart city and healthcare system to the scenario in Figure 4 COVID-19 Scenario: Outbreak of COVID-19 in Korea, we arrived at a smart city service structure for countering pandemics, presented in Figure 7.

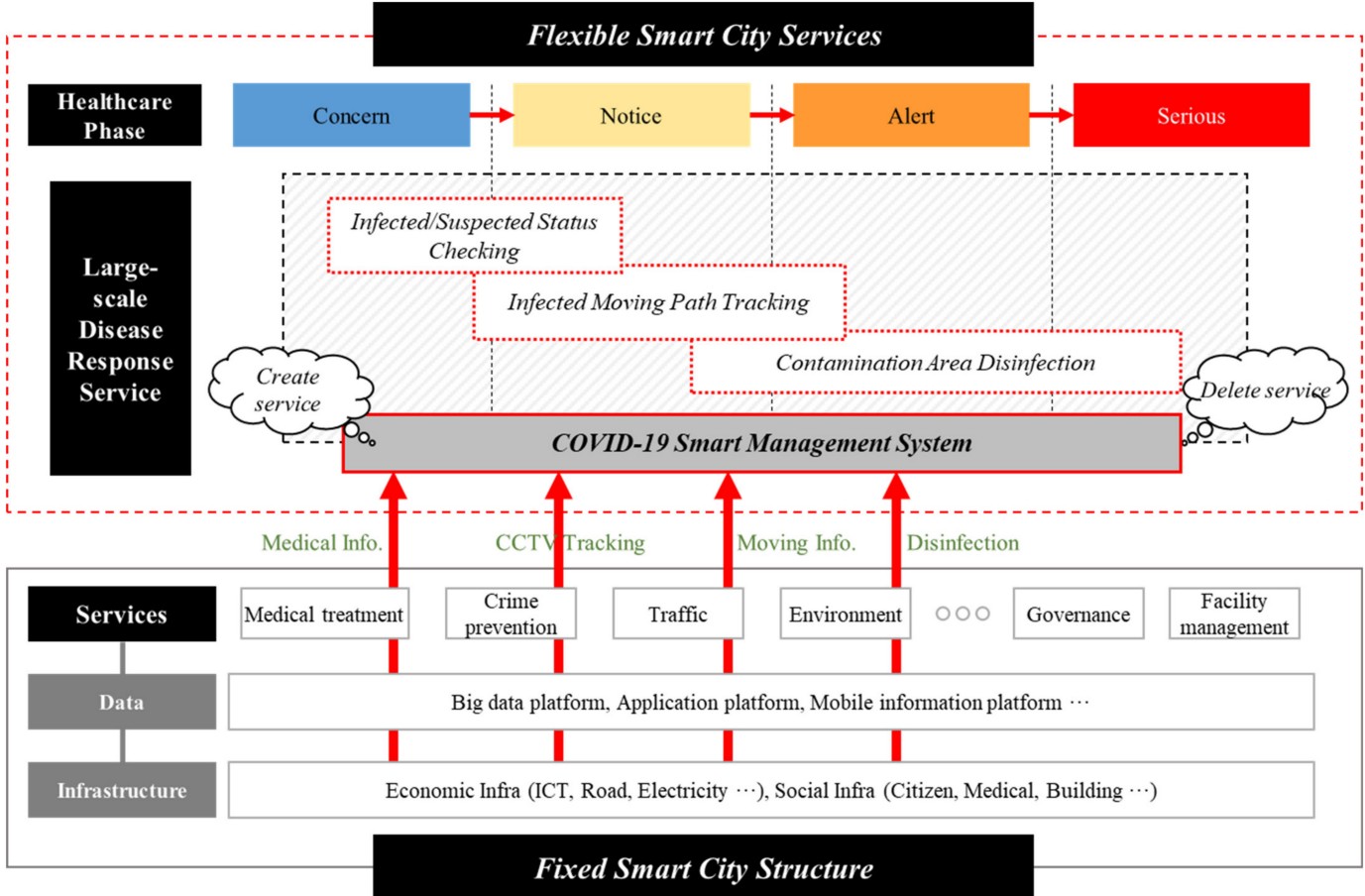

**Figure 7.** Smart city service structure for countering pandemics.

The result of contextualizing the scenario in Figure 4. COVID-19 Scenario: Outbreak of COVID-19 in Korea can be described as follows. Of all the smart city services, medical treatment services are utilized during the concern phase. This enables monitoring people suspected of having COVID-19, and analyzing the increase in numbers, to determine whether the pertaining situation could develop into a pandemic.

During the notice phase, a confirmed patient's movement path is analyzed on the basis of information received from the medical treatment service. Additionally, disaster prevention services are utilized along these paths of movement. The crime prevention service verifies these paths through CCTVs, and the traffic service confirms the public transportation routes of the confirmed patient. This verified information is shared with the environment service for disinfection of the concerned routes. In the meantime, the crime prevention service further identifies unconfirmed patients, or people with suspected symptoms, through thermal CCTVs. Moreover, the traffic service outlines routes for transporting emergency patients, while the medical treatment service provides the locations of available hospitals.

The smart city services utilized in the concern and notice phases were continued in the alert and serious phases for slowing down the spread of the disease. Thus, tracking information on confirmed patients and people in close contact with them continued, along with disinfection services.

The expanded smart city structure for countering pandemics was built on the pre-existing structure, and because of the urgency of the COVID-19 situation, no changes were made to the earlier definition, establishment and operation. As already mentioned, the integration of smart city services with the healthcare system involves a possible breach of personal information. Therefore, such smart city services should be offered only during crises such as a pandemic. In other words, these services should be made available when required and discontinued when the crisis ends.

## 4. Conclusions

Ever since the outbreak of COVID-19 in 2020, strategies have been continuously suggested to counter its quick spread in urban areas all over the world. Among the several proposed solutions, certain methods use IT to predict the outbreak of infectious diseases, some track transmission routes, while others aim at preventing disasters. However, smart city services, which the global community is keen on implementing, have been little studied, particularly with regard to countering the transmission of infectious diseases. Hence, we presented phases of countering outbreaks and transmission of infectious diseases, through the integration of smart city services with the healthcare system, for time-series analyses.

This study can be summarized as follows. First, existing smart city services are incapable of preventing or predicting new outbreaks of pandemics. This is due to the inflexible service structure of smart cities. To prove this postulate, this study assumed a scenario, as Figure 4. COVID-19 Scenario: Outbreak of COVID-19 in Korea, based on the COVID-19 outbreak in Korea from February to December 2019, in combination with the healthcare system for countering COVID-19. The study then contextualized the smart city structure in this scenario. The results show that all the smart city services could not provide essential data or solutions for countering COVID-19, especially in the early phases of the outbreak.

Second, it is important to realize that the suggested smart city services should only be offered during crises, such as a pandemic. Highly infectious diseases such as COVID-19 require the classification of infected persons and the identification of their paths of movement to slow down or prevent spread in urban areas. Simply put, these services use personal information, as they are required to track patients' paths of movement for curbing the spread of the disease. Thus, such services should be scalable, that is, made available in a special circumstance and rendered unavailable when circumstances change. In Korea, disease-related smart city services are legally sanctioned to be used only when necessary to protect personal information.

This study also has limitations. First, it did not consider privacy violations in the premise. Smart city technology utilizes personal information (e.g., credit card and public transportation usage) to track the transmission routes of infectious diseases. Breach of personal information is a sensitive issue; therefore, the discussion on whether public interests or personal privacy needs prioritization continues.

Second, the Figure 4. COVID-19 Scenario: Outbreak of COVID-19 in Korea scenario assumed by this study is uncertain. The scenario further assumes that all the confirmed COVID-19 patients were identified. However, it is practically impossible to sort out all COVID-19 patients or detect all possible cases that have not yet been identified. As this problem pertains to neither the healthcare system nor smart city technology, it was not covered in this study. In addition, epidemics in cities are not spread simply by one factor. It is important to operate the monitoring system for sewage and wastewater for population-level assessment of infection, which is considered to be the most important source of infection. Therefore, they should be considered in future research.

It is possible to devise strategies that can counter future pandemics by converging smart city services with other existing services. However, smart city is a public service, and the expanded application of its services could worsen privacy violations, although it might effectively control urban problems. This had been a controversial issue since even before

the concept of smart city emerged. Nonetheless, concerns of "Big Brother" (surveillance) should be considered gravely, as smart city technology continues expanding.

**Author Contributions:** Conceptualization, J.-H.K.; methodology, J.-H.K.; validation, J.-H.K. and J.-Y.K.; formal analysis, J.-H.K.; investigation, J.-H.K. and J.-Y.K.; resources, J.-H.K. and J.-Y.K.; data curation, J.-H.K.; writing—original draft preparation, J.-H.K.; writing—review and editing, J.-H.K. and J.-Y.K.; visualization, J.-H.K. and J.-Y.K. All authors have read and agreed to the published version of the manuscript.

**Funding:** This research received no external funding.

**Institutional Review Board Statement:** Not applicable.

**Informed Consent Statement:** Not applicable.

**Data Availability Statement:** Not applicable.

**Conflicts of Interest:** The authors declare no conflict of interest.

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
