# Peer review of "How Should the Structure of Smart Cities Change to Predict and Overcome a Pandemic?"

_sustainability, doi:10.3390/su14052981_

Round 1
Reviewer 1 Report
So, I like this article. I think it synthesizes some important and useful literatures, and has some useful points to make. That said, I think it requires some tweaks to make it as effective as it could be. Broadly, those are as follows:
1) The abstract is very confusingly written and needs a lot of editing. This is the place where the story must be told as simply and clearly as possible, and I think a lot of the abstract is confusing and hard to interpret.
2) I have some theoretical questions about the applicability of these approaches outside of Korea. For technical reasons (most smart cities have less capability and are less developed than Korean ones), but especially for legal and normative reasons (concerns about surveillance, governance, cybersecurity, etc.) I think many cities and countries will not be comfortable with this approach. I think this needs some exploration.
3) I'd argue that one of the areas in which Smart City applications (in particular) and urban automation (in general) have been most important during the COVID pandemic - the monitoring of sewage and wastewater for population level assessment of infection - isn't even mentioned. I think that is both a big oversight, and an opportunity for critics. I think that piece needs to be included and worked into any assessment of smart cities and pandemics. Additionally, I've included a few other pieces on COVID and smart cities.
1) Confusing or awkward wording in the abstract:
“Additionally, smart city services couldn`t respond in-time series” (is this time series in the statistical sense? Just meaning sequential?)
“a method responding in-time sequential by flexibly combining”
“Second, flexible smart city services are combined and deleted as needed to cope with COVID-19 has 17 been summarized. Third, smart city services should only be used to cope with pandemic situations: 18 the healthcare-system data consists of personal information. Therefore, smart city services respond- 19 ing COVID-19 must exist as a flexible.”
2) Theoretical/Substantive questions:
- (pg2) Does the use of police/telecom monitoring of patients translate out of the Korean context? (i.e. would it be constrained by laws and norms elsewhere?) And what are the implications of this.
- (pg3) Tables 1 is interesting, but it’s not clear the source of the data. Is this from the literature? (if so, needs sourcing), or a synthesis by the authors? This may be because of the error message in sourcing in the document I received.
- (pg4) Table 3 isn’t clear to me. What is “EA”? Not spelled out.
- (pg5) Table 4 is very useful.
- (pg14) “such services should be flexibly used only during a national crisis and removed when the service is no longer needed.” Is this likely to happen? Are there concrete examples of such systems being built and operated, then just turned off? Who would oppose turning them off? (I’d imagine many stakeholders)
3) Key missing components/literatures:
Smart City and COVID
Webb, W., & Toh, C. K. (2020). The smart city and COVID‐19. IET Smart cities, 2(2), 56-57.
Inn, T. L. (2020). Smart city technologies take on COVID-19. World Health, 841.
Sharifi, A., Khavarian-Garmsir, A. R., & Kummitha, R. K. R. (2021). Contributions of smart city solutions and technologies to resilience against the COVID-19 pandemic: a literature review. Sustainability, 13(14), 8018.
Gusikhin, O. (2020). The Impact of COVID-19 Experience on Smart City and Future Mobility. In Smart Cities, Green Technologies, and Intelligent Transport Systems (pp. 308-321). Springer, Cham.
Kim, H. M. (2021). Smart cities beyond COVID-19. In Smart Cities for Technological and Social Innovation (pp. 299-308). Academic Press.
Monitoring of sewage/wastewater for COVID case estimation
Farkas, K., Hillary, L. S., Malham, S. K., McDonald, J. E., & Jones, D. L. (2020). Wastewater and public health: the potential of wastewater surveillance for monitoring COVID-19. Current Opinion in Environmental Science & Health, 17, 14-20.
Daughton, C. G. (2020). Wastewater surveillance for population-wide Covid-19: the present and future. Science of the Total Environment, 736, 139631.
Bogler, A., Packman, A., Furman, A., Gross, A., Kushmaro, A., Ronen, A., ... & Bar-Zeev, E. (2020). Rethinking wastewater risks and monitoring in light of the COVID-19 pandemic. Nature Sustainability, 3(12), 981-990.
Jaiswal, R., Agarwal, A., & Negi, R. (2020). Smart solution for reducing the COVID‐19 risk using smart city technology. IET Smart Cities, 2(2), 82-88.
Reviewer 2 Report
This article advocates for better flexibility (in the sense of adding and removing when necessary) of smart city services and better integration between them. The authors argue that this would be precious to predict and fight against a pandemic, and as such they propose a smart city structure model.
First, I have a series of concerns regarding the form. There are some phrasing issues and typos throughout the paper. There are also many in-text reference errors with tables, which makes it hard to follow given the high number of tables (13!). Also, references to figures seem to include the full caption and cause line breaks instead of the figure number only. Please check especially the conclusion, this is hardly readable. There are referencing issues with sections as well (e.g. line 213: “As mentioned in 0”) Finally, and most importantly, the format of the bibliography is highly problematic. For many references (e.g. [15], [22], [25]) it is impossible to retrieve the source. For many other sources, the local access path of the file is provided, which again prevents the reader from retrieving the source. All these issues could have been picked up by a simple check. Unfortunately, they are enough for me to recommend to reject the paper, as I consider that submitting a report of scientific quality in this form is unacceptable.
The content of the paper raises major concerns as well.
In the introduction, I could not understand the motivation of the paper. The authors argue that the smart city services failed to be flexible enough to predict and fight the pandemic. However, I am lacking a concrete example of failure that proves the authors’ point. It is well-known that different services are hard to integrate together (this is not specific to the smart city), but I would have liked the authors to illustrate in light of the pandemic context, and maybe to the context of Korea since their contribution is specific to this geographic context in that it integrates the 12 smart city categories of Korea and the healthcare phases of Korea. The contribution is not presented in the introduction, only the problem is.
The literature review is very long (it represents half of the paper) and can be significantly shortened as it contains few information that is actually useful to the rest of the paper. It is also hard to read, as it contains more tabular content than text. In Table 3, what is an EA? Table 4 presents smart city projects of several different countries. The authors than write that these services have a limitation in that cannot predict and prevent transmission of threats such as COVID. It the threats concern COVID only, it does not really make sense, as many of the mentioned services simply do not have this goal (e.g. smart grid). If it goes beyond COVID, this is a claim that needs to be backed with a sound and transparent evaluation by the authors or literature. Table 5 is useless in my opinion. It is well-known that many standards exist and are not used in every service, which causes integration issues. The authors could remove the table and mention 1-2 standards relevant for pandemics and make the same point. Section 2.3 has the same title as Section 2.2. It presents phased plans for pandemics. Several different plans are described, which is useless information for the reader. Indeed, the phases in a Texan county are of no use to understand the authors’ contribution. They could easily break the section down to the essential information that would be the WHO phases and the Korean healthcare phases, and mention that there are global and local plans throughout the world with different phases. Tables 8 and 9 can thus be removed. Table 7 is not very informative, all the phases say that actions from the previous phase should be continued or initiated. I have checked the online source and it mentions other aspects such as treatment and isolation for the recognition phase, which have not been picked up by the authors. A case study approach restricted to Korea would be more informative and more consistent with the contribution of the paper, which is as I mentioned specific to Korea.
Section 3, named “Main discussion”, should be renamed to be more explicit regarding what the reader can expect to find. I suggest e.g. “Revised model of smart city service structure”.
The contribution is a proposition of services such as screening people with thermal cameras and tracking infected people through CCTV. However, it misses a critical point of COVID. Infected people are in the public space without knowing that they have COVID, since contagion happens before symptoms appears. Once infected people have been identified, they are supposedly in quarantine. Do the authors propose to track everyone and then go back to the records of infected people to retrieve their paths and contacts? Or do the authors want to check if infected people are outside their home instead of in quarantine? This is not clear to me. Also, and more importantly, the integration and flexibility aspects are not really addressed in the end. The authors do not explain how the services they propose could be integrated with existing ones (e.g. which standards should be used) and how the services can be added or removed. This part is only represented as clouds in the proposed structure model but not discussed further. Therefore, it seems that the contribution breaks down to proposing strict tracking services which feasibility and acceptance by the population is not assessed.
My last concern is the ethical aspects of the proposed services. The authors acknowledge that not addressing these is a limitation of their research. In my opinion, this is more than a limitation, this is essential to discuss given the nature of what the authors propose.
In summary, my suggestions are to focus the paper on the Korean context, in an in-depth case study approach where authors detail more their contribution, clearly illustrate what the problem is and how their contribution helps solving it.
Reviewer 3 Report
The manuscript is quite interesting and widely fine. However, please include following melioration advices:
- Please always write out the full terms and than in brackets the abbreviation (lines 79-80)
- The whole manuscript is full of "Error! reference source..." (e.g. line 85)
- Line 142: lack of flexibility
- Please always explain abbreviations before use - e.g. line 149
- Line 320: according to the introduction the COVID-19 pandemic started in 2019 and here you state 2020
- Lines 352-354: here we have two times Figure 2. Moreover, please bring the text together.
- In this source you can find a whole list of smart city technologies http://www.sinfonia-smartcities.eu/en/resources/d21--swot-analysis-report-of-the-refined-conceptbaseline Might be of interest for your work
Round 2
Reviewer 1 Report
I think the paper, and especially the abstract, are much improved. Both in terms of readability and accessibility, but also in terms of context and framing.
I think the paper has the ability to make a contribution to the literature in a way that the previous version may not have been able to.
I think mentioning the scenario approach in the abstract might be worthwhile, but I don't think it's a huge problem if it isn't mentioned. Either way, I think the paper is in much better shape.
Reviewer 2 Report
The authors made substantial efforts to take my comments into account, and I commend them for that. However, despite the good improvements, I have some remaining concerns.
There are still some "Reference not found" errors and line breaks when referring to figures in the text, but almost everything regarding the format has been fixed, which gives the revised paper a much more professional look than its previous version. Regarding the references, my point was that there are many references *with formatting mistakes in them*, and not that there are too many references in the bibliography. Please feel free to add them back if you feel they should be included.
The motivation of the paper is still unclear for me in the introduction, and the reasons for the limitations of smart city services are not clear in Section 2, but is much better explained later in the paper, which is a nice improvement. I believe that explaining standard constraints (due to the multiplicity of service providers) and coupling between services in the introduction to motivate the paper would be valuable. Indeed, if my understanding is correct, they are the factors that cause the limitations of smart city services the authors want to address.
The authors made some efforts to streamline Section 2, and it is much easier to follow in its current form. However some parts still appear unnecessary to me. Table 3 just shows that medical, crime prevention, transportation, and environmental services are the most frequent, which is already said in the text, but without an explanation of what pilot city, regulatory sandbox, and smart city challenge are, it is difficult to extract anything else from this table. Table 7 does not bring much useful insights as well in its current form. The authors have not responded to my previous comment on this table, I thus redirect them to my review of the previous version.
Section 3.1 gives an overview of the COVID-19 and smart city systems situation in South Korea, but is not part of the novel contribution by the authors. I would thus include it in Section 2 instead. Section 2 would thus have a part on smart city services, a part on COVID-19 response plans, and then the current content of 3.1 which deals with the relationship between the two parts of the current Section 2. It would allow a smoother transition between Section 2 and Section 3, as the modified Section 2 would conclude with the need for a novel smart city service structure.
From the additional explanations in the text, I understand that the idea is to track the movements of everyone and then use that information to trace back the contacts of confirmed infected patients. I stand by my position that the feasibility/ethical aspects should be discussed, at least from a social acceptance or a legal point of view, as the reader is left skeptical about whether it can actually be done with the current explanations. The authors bring up interesting legal considerations in their response to my comments, it would be nice to have them in the paper to show the reader under which conditions the authors' solution can be used.
Finally, and this is my biggest remaining concern, the authors have not responded to my comment regarding the lack of explanations on how the flexible adding and removal of the proposed services would be achieved. This is still unclear to me in the paper. This is a very important point, since it changes what is the contribution of the paper. The current contribution seems to be the proposal of new services to handle the COVID situation, or similar situations in the future. However, I am lacking the contribution on the more general issue of integrating and removing services in an existing smart city infrastructure. In other terms, what can we use from the authors' paper to achieve a more flexible smart city service structure?
In brief, the authors have made major improvements on the form and good improvements to the content. However, there are still several of my previous comments that were not answered or not answered in a sufficiently convincing way.
Round 3
Reviewer 2 Report
Regarding the form, the reference errors have been removed, but the line breaks when referring to figures are still there. I will give a clearer example from p.15, line 345:
Second, the (
Figure 4. COVID-19 Scenario: Outbreak of COVID-19 in Korea ) scenario assumed by this study is uncertain.
should appear as
Second, the Figure 4 scenario assumed by this study is uncertain.
Regarding the content, the motivation is clearer in the abstract, and this clarifies the contribution as well. I thus find it now less essential to discuss the practical aspects of service adding and removing. The legal and ethical aspects are still too much overlooked in my opinion, but I appreciate that the authors added a note on the legal framework of Korea. I still encourage the authors to have a deep reflection on the legal and population acceptance of their proposal, but this can be the goal of another paper. I was not conviced by the authors' response regarding my suggestion for putting 3.1 into Section 2, but this was a quite minor comment anyway, and there is not only one good way to structure a text, the paper is also understandable with the structure proposed by the authors.
In brief, the authors' changes and explanations are a bit light for what I expected from a major revisions recommendations, but they added some nice clarifications that, for the most part, satisfy my concerns. So I am fine with the article being published.